# M²SODAI: Multi-Modal Maritime Object Detection Dataset With RGB and Hyperspectral Image Sensors

**Jonggyu Jang**
POSTECH
jgjang@postech.ac.kr

**Sangwoo Oh**
KRISO
swoh@kriso.re.kr

**Youjin Kim**
Samsung Electronics
youjin8022@gmail.com

**Dongmin Seo**
Semyung University
dseo@semyung.ac.kr

**Youngchol Choi**
KRISO
ycchoi@kriso.re.kr

**Hyun Jong Yang**[*]
POSTECH
hyunyang@postech.ac.kr

## Abstract

Object detection in aerial images is a growing area of research, with maritime object detection being a particularly important task for reliable surveillance, monitoring, and active rescuing. Notwithstanding astonishing advances in computer vision technologies, detecting ships and floating matters in these images is challenging due to factors such as object distance. What makes it worse is pervasive sea surface effects such as sunlight reflection, wind, and waves. Hyperspectral image (HSI) sensors, providing more than 100 channels in wavelengths of visible and near-infrared, can extract intrinsic information about materials from a few pixels of HSIs. The advent of HSI sensors motivates us to leverage HSIs to circumvent false positives due to the sea surface effects. Unfortunately, there are few public HSI datasets due to the high cost and labor involved in collecting them, hindering object detection research based on HSIs. We have collected and annotated a new dataset called "**Multi-Modal Ship and flOating matter Detection in Aerial Images (M²SODAI)**", which includes synchronized image pairs of RGB and HSI data, along with bounding box labels for 5,764 instances per category. We also propose a new multi-modal extension of the feature pyramid network called DoubleFPN. Extensive experiments on our benchmark demonstrate that the fusion of RGB and HSI data can enhance mAP, especially in the presence of the sea surface effects. The source code and dataset are available on the project page: https://sites.google.com/view/m2sodai.

## 1 Introduction

With the growing maritime traffic intensity, detecting and localizing ships and floating matters have become core functionalities for reliable monitoring, surveillance, and active rescuing [3, 7]. Conventionally, there have been sea surface maritime surveillance systems based on buoys and ships [56]. These systems are cost-efficient; however, their sensing range is relatively narrow. By virtue of their wide sensing range, aerial surveillance systems have received considerable research interest, the absolute majority of which leverage optical cameras. Although optical cameras can obtain high-resolution RGB images, the competence of optical sensors is degraded under dire but commonplace environmental conditions such as solar reflection or waves, *i.e.*, *sea surface effects*.

Hyperspectral image (HSI) sensors, which acquire imagery in hundreds of contiguous spectral bands, are emerging as a substitute or supplement of RGB sensors [42, 32]. Abundant spatio-spectral

---

[*]corresponding author

37th Conference on Neural Information Processing Systems (NeurIPS 2023) Track on Datasets and Benchmarks.

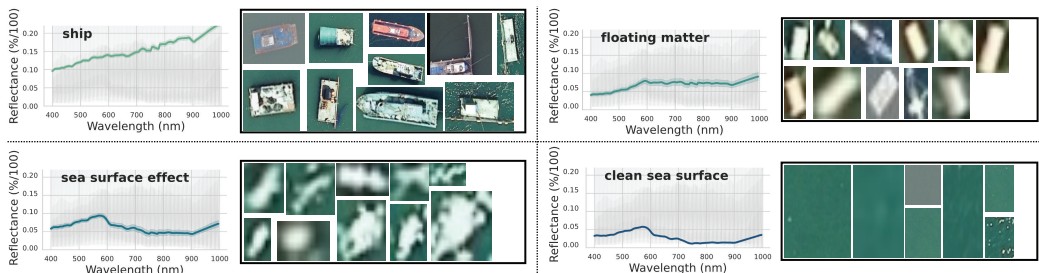

Figure 1: M$^2$SODAI dataset spectral analysis. From the top of the figure, we depict the hyperspectral reflectance intensity patterns and cropped RGB data of i) ship, ii) floating matter, iii) sea surface effect, and iv) clean sea surface. The figure shows that the floating matters and sea surface effects are similar in the RGB image; however, they have different reflectance intensity patterns in the HSI data.

Table 1: M$^2$SODAI dataset vs. related datasets for RGB and HSI data. Among all the datasets, the M$^2$SODAI dataset is the only dataset with i) bounding-box-annotated, ii) synchronized multi-modal, and iii) aerial RGB and HSI data.

| dataset | #instances /class | #images | #classes | RGB data (width) | HSI data (width) | multi-modality | annotation | view | Year | description |
|---|---|---|---|---|---|---|---|---|---|---|
| VEDAI[38] | 327 | 1,268 | 9 | ✓ (512, 1,024) | - | - | bounding box | aerial | 2015 | object detection |
| COWC[34] | 2,007 | 32,700 | 1 | ✓ (2,048) | - | - | bounding box | aerial | 2016 | vehicle detection |
| CARPK[21] | 89,777 | 1,448 | 1 | ✓ (1,280) | - | - | bounding box | aerial | 2017 | vehicle detection |
| DOTA-v1.0[50] | 12,552 | 2,806 | 15 | ✓ (800-13,000) | - | - | bounding box | aerial | 2018 | object detection |
| VisDrone[59] | 5,420 | 10,209 | 10 | ✓ (2,000) | - | - | bounding box | aerial | 2018 | object detection |
| iSAID[49] | 43,696 | 2,806 | 15 | ✓ (800-13,000) | - | - | polygon | aerial | 2019 | object detection |
| FGSD[7] | 131 | 2,612 | 43 | ✓ (930) | - | - | bounding box | aerial | 2020 | ship detection |
| DOTA-v2.0[8] | 99,647 | 11,268 | 18 | ✓ (800-20,000) | - | - | bounding box | aerial | 2021 | object detection |
| India Pines[2] | - | 1 | 16 | - | ✓ (145) | - | pixel-wise | aerial | 2015 | remote sensing |
| HAI[31] | - | 65,000 | - | ✓ (500) | ✓ (500) | ✓ (Sync) | - | aerial | 2021 | dehazing |
| Samson[58] | - | 1 | 3 | - | ✓ (952) | - | pixel-wise | aerial | 2022 | remote sensing |
| MDAS[22] | - | 23 | 859 | ✓ (15,000) | ✓ (300) | ✓ (Sync) | pixel-wise | aerial | 2022 | remote sensing |
| HS-SOD[23] | 120 | 60 | 1 | - | ✓ (1,024) | - | polygon | terrestrial | 2018 | object detection |
| ODHI[52] | 832 (RGB), 207 (HSI) | 2048 (RGB), 454 (HSI) | 8 | ✓ (∼696) | ✓ (∼696) | × (Async) | bounding box | terrestrial | 2021 | real/fake detection |
| **M$^2$SODAI (ours)** | **5,764** | **1,257** | **2** | **✓ (1,600)** | **✓ (224)** | **✓ (Sync)** | **bounding box** | **aerial** | **2023** | **object detection** |

snapshots of HSIs provide inherent reflective properties of materials even with just a few pixels, which is not possible with RGB or any other types of images. Figure 1 shows the RGB and HSI data examples of the ships, floating matters, sea surface effects, and clean sea surface, where reflection intensity patterns of objects and backgrounds are plotted in the left parts. In the wavelengths of the near-infrared (NIR) region, water exhibits a pattern of sharply decreasing reflectance between 700 nm and 900 nm, unlike target objects [54]. That is, even at low resolution, HSI sensor data can identify unique object characteristics, differentiating the targets from the background[2].

However, most of the object detection datasets on aerial images are about optical images [7, 38, 34, 21, 50, 49, 8], and there are only handful HSI datasets publicly available. Even for other tasks, such as remote sensing, datasets with aerial HSIs are scarce because collecting HSI data is costly and labor-intensive [2, 58, 22]. In this work, we build a new Multi-Modal Ship and flOating matter Detection in Aerial Images (M$^2$SODAI) dataset, which contains synchronized pairs of aerial RGB and HSI data. For the data collection, we used an off-the-shelf HSI sensor taking 127 spatio-spectral channels for each snapshot on the wavelength from 400 nm to 1000 nm in steps of 4.5 nm. The spatial resolutions of the RGB and HSI data are 0.1 m and 0.7 m, respectively, at the altitude of 1 km.

For object detection in aerial images, one major drawback of HSIs is their relatively low spatial resolution (several meters) compared with optical images (tens of centimeters). Thus, HSI sensors have been commonly used in remote sensing systems which do not require high-resolution [14, 13, 1, 25]. As hardware technologies for HSI sensors evolve, their resolution has increased, facilitating

---

[2]For a detailed analysis, please see Appendix E.

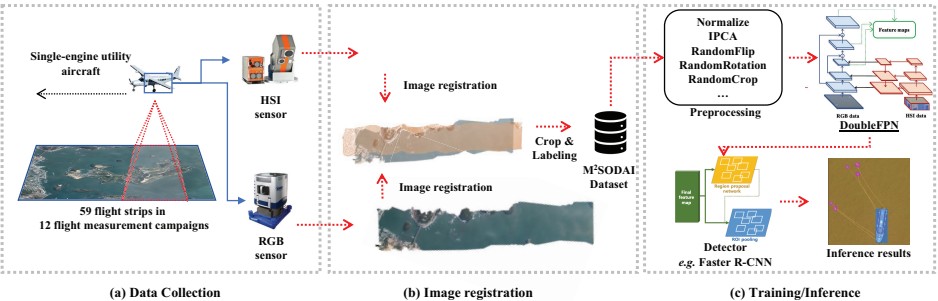

**(a) Data Collection**   **(b) Image registration**   **(c) Training/Inference**

Figure 3: Illustration of the overall procedure of our work. (a) We collect the sensor data using a single-engine utility aircraft equipped with RGB and HSI sensors. (b) Due to the offset between RGB and HSI sensor data, we register RGB and HSI data into the same coordinate and then construct our dataset by cropping and labeling data. (c) The data are preprocessed and forwarded to the DoubleFPN layer. Finally, after the DoubleFPN layer, the detector estimates the bounding boxes of the target objects from the DoubleFPN output.

bounding-box-based deep learning research for object detection in HSIs [52]. Nonetheless, the resolution of off-the-shelf HSI sensors is not yet high enough for airborne surveillance systems. Therefore, it is a better choice to use HSIs as a supplement, not a substitute, to optical images in the case of far-field object detection. Further analysis of related works is provided in Appendix A.

The salient contributions of our work are listed as follows:

- **M²SODAI dataset:** The M²SODAI dataset is the first multi-modal, bounding-box-labeled, and synchronized aerial dataset, featured by 11,527 instances, 1,257 images, and synchronized RGB-HSI data. In Tab. 1, we compare the M²SODAI dataset with the related public datasets on RGB and HSI data. Amongst the related datasets, the HAI [31] and MDAS [22] datasets only provide synchronized multi-modal aerial data; however, they come with no annotation [31] or low-resolution pixel-wise annotation [22].

    - **Raw data processing:** We add a contrast enhancer to the method proposed in [24] for more accurate data synchronization of RGB and HSI. For more details, please refer to Sec. 2 and Appendix D. Figure 2 illustrates randomly selected pairs of RGB and HSI data from our dataset.

- **Multi-modal benchmark and learning framework:** We conducted an object detection benchmark with our dataset, where the graphical and numerical results ensure the HSI data can enhance the detection accuracy, especially for data with sea surface effects. In order to fuse the RGB and HSI data, we propose an extension of the feature pyramid network (FPN) [26], DoubleFPN. For a more general benchmark, we build other fusion methods based on DetFusion [44] and UA-CMDet [43].

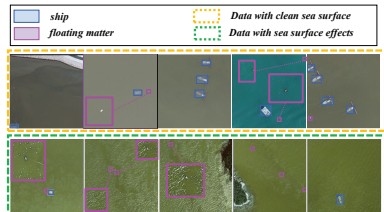

(a) Annotated RGB data.

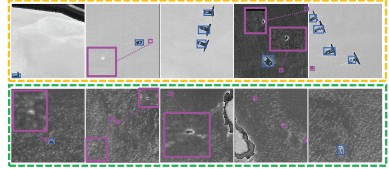

(b) Annotated HSI data.

Figure 2: Examples of collected RGB and HSI data. (a): Typical RGB data in M²SODAI dataset. (b): Infrared visualization of the synchronized HSI data in the dataset (ratio of the 25-th and 72-nd channels). Here, we show ten examples of images with and without sea surface effects. In the images with sea surface effects, the HSI data have more recognizable features than the RGB data.

## 2   M²SODAI Dataset

**Overview of the proposed dataset construction procedure**   Figure 3 illustrates the overall procedure of the proposed scheme. In the first stage, we collect RGB and HSI sensor data using a single-engine utility aircraft equipped with RGB and HSI sensors. In the second stage, an image

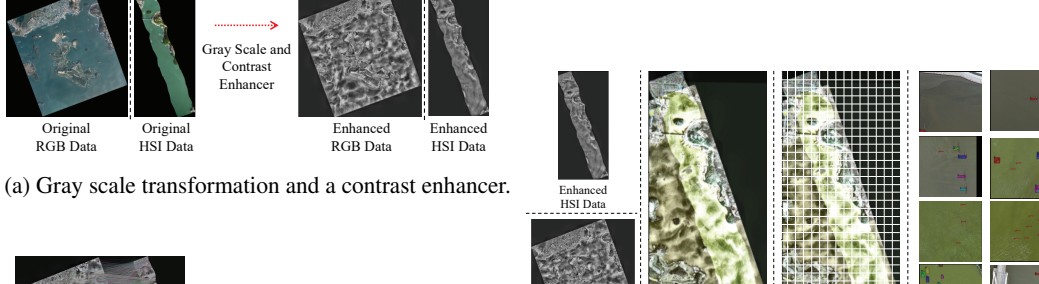

(a) Gray scale transformation and a contrast enhancer.

$$H = \begin{bmatrix} 1.02 \cdot 10^0 & -2.87 \cdot 10^{-2} & -4.47 \cdot 10^3 \\ 1.65 \cdot 10^{-2} & 1.00 \cdot 10^0 & 2.74 \cdot 10^2 \\ 6.47 \cdot 10^{-7} & -4.51 \cdot 10^{-7} & 1.00 \cdot 10^0 \end{bmatrix}$$

ORB Feature Extractor and Brute-Force Matcher

Homography Matirx

(c) Image registration and dataset construction.

(b) Feature matching and corresponding homography matrix.

Figure 4: Illustration of our dataset construction procedure. (a) The RGB and HSI sensor data are transformed by a gray scaler and a contrast enhancer. (b) The matched feature of transformed sensor data and the corresponding homography matrix are obtained. (c) The sensor data are registered, cropped, and labeled.

registration method is used to coincide the pixels of RGB and HSI data. After the image registration, we construct our dataset by cropping by fixed size and annotating target objects (ships and floating matters) in the RGB and HSI data. Note that our dataset consists of HSI, RGB, and corresponding bounding box annotation data. Further details of our dataset are available in Appendix B. In the third stage, we train our DoubleFPN architecture and evaluate the trained model using the M$^2$SODAI dataset.

**Data collection** Our focus is to create a public dataset consisting of synchronized maritime aerial RGB and HSI data. To this end, we built a data collection system by leveraging a single-engine utility aircraft (Cessna Grand Caravan 208B). An HSI sensor (AsiaFENIX, Specim, Oulu, Finland) and an RGB sensor (DMC, Z/I Imaging, Aalen, Germany) are equipped on the bottom of the aircraft, the direction of which is downward. The raw data was acquired through 59 flight strips in 12 flight measurement campaigns, which cover a total area of 299.7 km$^2$. During the flight strips, the aircraft maintains its speed of 260 km/h and altitude of 1 km.

Table 2: Specification of RGB and HSI sensor. The resolutions of the sensors are corresponding to the aircraft's altitude of 1 km.

|  | HSI Sensor | RGB Sensor |
|---|---|---|
| Name | AisaFENIX (@Specim) | DMC (@Z/I Imaging) |
| Spectrum | 400-1000 nm (in steps of 4.5 nm) 127 channels | Blue: 400-580 nm Green: 500-650 nm Red: 590-675 nm |
| Altitude | 1 km | |
| Field of View | 40° | 74° |
| Resolution | 0.7 m | 0.1 m |

Table 2 shows the detailed specifications of the sensors used in the data collection. The HSI sensor (AsiaFENIX) scans the wavelength range from 400 nm to 1000 nm in steps of 4.5 nm, a total of 127 spectrum bands. The wavelength range includes visible spectrum and NIR spectrum, generally used for remote sensing and machine vision tasks. The RGB sensor (DMC) captures high-resolution RGB data in three channels: Red (590-675 nm), Green (500-650 nm), and Blue (400-580 nm). We note that RGB and HSI data are collected simultaneously, in which the spatial resolutions of RGB and HSI sensors are approximately 0.1 m and 0.7 m, respectively.

**Image registration and annotation** In the previous step, we introduce the methodology of collecting the raw RGB and HSI data. Since the size of the raw data is too large for object detection (HSI: 3,220$^2$ pixels, and RGB: 22,520$^2$ pixels on average), we cropped the raw data into a fixed size. We note that RGB and HSI data are cropped in size of $1600 \times 1600 \times 3$ and $224 \times 224 \times 127$, respectively. However, the problem is that the coordinates of the collected RGB and HSI pairs are not matched. Hence, we employ an image registration method to correct pixel offsets between RGB and HSI pairs. In Fig. 4, our data processing procedure is depicted.

1. We transform the raw RGB and HSI data into grayscale images (Fig. 4a) [24].

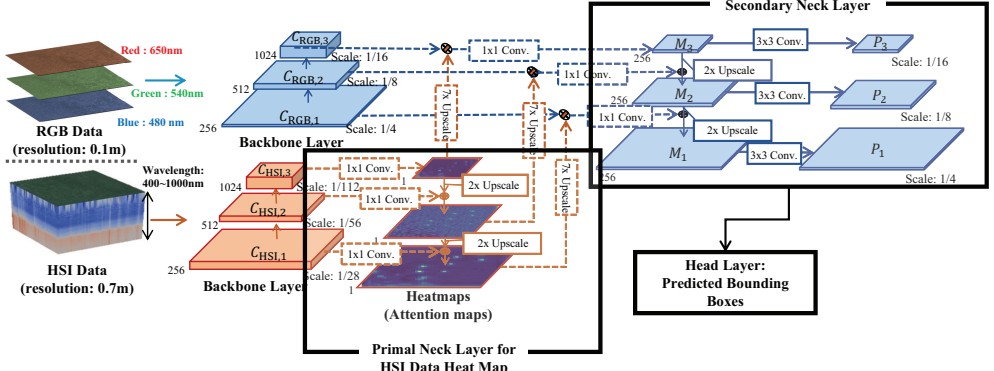

Figure 5: Schematic diagram of the DoubleFPN-based object detection architecture. The DoubleFPN object detection architecture consists of three sub-architectures: backbone, neck, and head layers. In the backbone layer, the feature maps of each input data are extracted, *i.e.*, bottom-up pathway. In the neck layer, the DoubleFPN fuses feature maps, *i.e.*, top-down pathway. In the head layer, the object detector estimates the classes and bounding boxes of the objects.

2. We apply contrast-limited adaptive histogram equalization (CLAHE)-based contrast enhancer to the grayscale RGB data and grayscale HSI data (Fig. 4a).

3. To estimate the homography matrix between the enhanced RGB data and enhanced HSI data, we carry out the oriented FAST and rotated BRIEF (ORB) feature descriptor [41] to both data, thereby extracting features of the data (Fig. 4b).

4. We use a Brute-force matcher to find the matched feature among the ORB features; then, the homography matrix is computed from least square optimization for synchronizing the matched features.

5. We crop the registered data in the same size and generate corresponding bounding box annotation data (Fig. 4c).

For object detection, we annotated the bounding boxes on the instances of two classes: 1) floating matter and 2) ships in both RGB and HSI data. We note that the following instances are labeled as floating matters: buoys, rescue tubes, small lifeboats, surfboards, and humans (mannequins[3]) with life vests. Also, for the ship class, we annotated bounding boxes on steamboats, cruise ships, fishing boats, sailboats, rafts, and other ship categories. We refer to the infrared visualization map of the HSI data for bounding box annotation. For labeling, two of the authors annotated target instances by using Labelme [47], in which the minimum size box containing each object was set as the policy, and multiple cross-checks were performed. For more details on raw data processing, please see Appendix D.

**Dataset splits**    After the data processing, we obtained 1,257 pairs of synchronized RGB and HSI data, where the total number of instances in the dataset is 11,527. For experiments, we randomly divided the dataset into 1,007 training data, 125 validation data, and 125 test data.

## 3 Method: DoubleFPN

The feature fusion methods are categorized into i) early fusion, ii) middle fusion, and iii) late fusion [11]. The early fusion methods fuse sensor data before the backbone layers, thereby fully leveraging joint features of raw data. However, the common representations of different sensor data are challenging. On the other hand, the late fusion methods combine sensor data just before the final detector, whereas they have a potential loss for finding the correlation of sensor data. In our study, the aim is a compromise proposal of early and late fusion methods, *i.e.*, *middle fusion*.

Here, the training/inference procedure in Fig. 3 is addressed. In the canonical FPN structure [40], a pyramid structure for feature extraction is proposed to resolve the issues of memory inefficiency and

---

[3]Distressing a real person was done with a mannequin for safety reasons.

Table 3: AP (%) benchmark result on the M$^2$SODAI dataset with the DoubleFPN and the uni-modal baseline methods. All the results are obtained by using ResNet-50 backbone and Faster R-CNN detector. In addition to the AP-based metrics, we show types of neck layers and use of the RGB and HSI data.

| neck layer | multi-modal | RGB data | HSI data | mAP | AP$_{@.5}$ | AP$_{@.75}$ | Ship | Float. Mat. | AP$_s$ | AP$_m$ | AP$_l$ |
|---|---|---|---|---|---|---|---|---|---|---|---|
| **DoubleFPN(ours)** | ✓ | ✓ | ✓ | **44.4** | **84.8** | 39.3 | 55.7 | **33.1** | **35.2** | 41.7 | **61.4** |
| FPN (RGB) [26] | × | ✓ | × | 38.8 | 77.0 | 33.3 | 52.4 | 25.2 | 18.3 | **44.8** | 55.6 |
| FPN (HSI) [26] | × | × | ✓ | 7.8 | 23.2 | 2.9 | 15.8 | 0.0 | - | - | - |
| UA-CMDet [43] | ✓ | ✓ | ✓ | 42.9 | 84.0 | **40.0** | **55.9** | 29.8 | 20.8 | 43.0 | 60.8 |
| DetFusion [44] | ✓ | ✓ | ✓ | 42.0 | 84.3 | 35.4 | 53.5 | 30.5 | 24.2 | 41.9 | 61.1 |
| Early fusion | ✓ | ✓ | ✓ | 42.9 | 83.0 | 37.6 | 54.2 | 31.5 | 18.9 | 44.1 | 59.7 |

*Best: **bold and underline**, second-best: underline.

low inference speed of the general feature map extraction architecture. However, the input of the FPN is a fixed-scale single image, and the output is feature maps sized proportionally to the input image.

For our dataset, the feature extraction network should be capable of handling RGB and HSI data with different scales. More importantly, HSI data itself does not have sufficiently high resolution to detect aerial objects, even though it can capture unique features of materials. Hence, we propose an extension of the canonical FPN to jointly extract feature maps by fusing two data. The detailed schematic diagram of the DoubleFPN is depicted in Fig. 5. We note that the DoubleFPN architecture can be generally implemented with other detectors, such as RetinaNet and FCOS [27, 45].

**Dimensionality reduction and preprocessing**    Let us denote the size of RGB data and HSI data as $H_{\text{rgb}} \times W_{\text{rgb}} \times 3$ and $H_{\text{hyp}} \times W_{\text{hyp}} \times C_{\text{hyp}}$, respectively. We note that $C_{\text{hyp}} = 127$ in our dataset. Since several spectral features are necessary for object detection, we leverage the incremental PCA method. As a result, we observe that the cumulative variance of the first 30 principal components occupies more than 99.9% of the total variance. Hence, we use 30 principal components in our object detection instead of fully leveraging 127 channels.

**Backbone layer**    In Fig. 5, the backbone layers are feed-forward CNNs that extract feature maps of the inputs, *i.e.*, *bottom-up pathway*. As in the figure, each pair of RGB and HSI data is fed into the separate backbone layer, in which the CNN layers for RGB and HSI data have $N$ different scales. The output feature maps at each level are scaled by 1/2 of that at the previous level. Here, we denote the $i$-th feature map of RGB and HSI data as $C_{\text{RGB},i}$ and $C_{\text{HSI},i}$.

**Neck layer**    In the neck layer, the DoubleFPN forwards $N$ fused feature maps from $N$ RGB feature maps and $N$ HSI feature maps. In the primal neck layer, the HSI feature maps are converted into attention maps to represent weights for the high-resolution RGB features. At the top of the primal neck layer, the feature map $C_{\text{HSI},N}$ is fed into $1 \times 1$ convolution layer with one channel with Sigmoid activation function. In the top-down pathway of the primal neck layer, the $i$-th feature map $C_{\text{HSI},i}$ is forwarded into $1 \times 1$ convolution layer with one channel and is added with the 2x up-scaled previous attention map. Let us define the $i$-th attention map as $H_i$. Then, the $i$-th attention map $H_i$ is up-scaled seven times and is multiplied with the $i$-th RGB feature map $C_{\text{RGB},i}$ for $i = 1, .., N$. In the secondary neck layer, the fused feature maps $H_i \cdot C_{\text{RGB},i}$ are forwarded into the canonical FPN structure. Consequently, after $3 \times 3$ convolution layers, we get a set of fused feature maps with different scales.

**Head layer**    The head layer predicts the bounding boxes and classes of the objects from the output of the DoubleFPN. For the experiments, we introduce an application of our method to Faster R-CNN [40]. For further implementation details, please refer to Appendix F.

## 4   Experiments

Since none of the other datasets in Tab. 1 provides synchronized RGB, HSI, and bounding-box-annotation data, we evaluate the DoubleFPN on the M$^2$SODAI dataset.

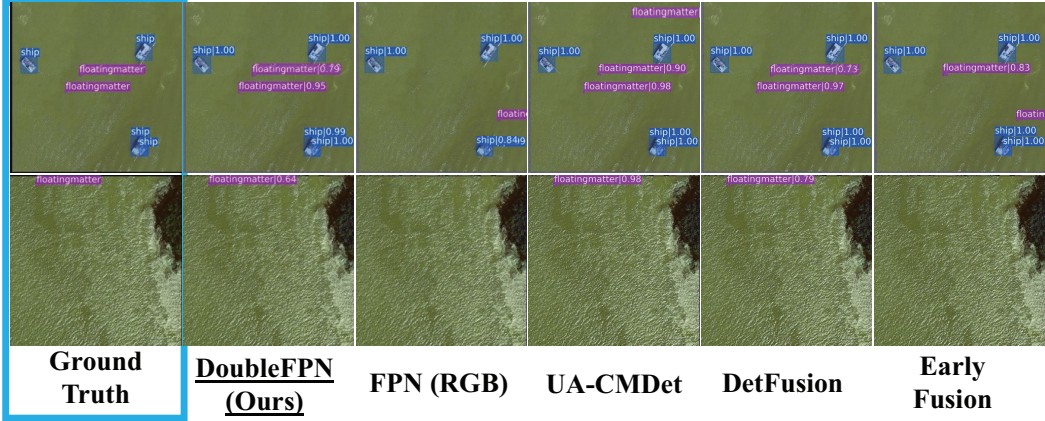

| Ground Truth | DoubleFPN (Ours) | FPN (RGB) | UA-CMDet | DetFusion | Early Fusion |

Figure 6: Detection results on data with sea surface effects. The first figure depicts the ground truth of the bounding box. The other figures show the detected bounding boxes of the objects, *e.g.*, floating matter, ship.

## 4.1 Setups and Implementation Details

**Implementation details** Our experiment is carried out on two NVIDIA RTX 3090 GPUs. The overall object detection model is trained for 73 epochs, in which the stochastic gradient descent parameters are: the learning rate of $2 \cdot 10^{-2}$, the momentum of $0.9$, and the weight decay of $1 \cdot 10^{-4}$. In addition, the batch size is set to be one per GPU[4]. For fairness in the performance analysis, we evaluate all methods based on the ResNet-50 backbone model [19]. Since the ResNet-50 model provides five-stage feature maps, each of the backbone networks in the uni-modal methods provides five feature maps. In the DoubleFPN, the backbone layer for RGB and HSI data input forwards five and four feature maps, respectively, *i.e.* $N = 5$. For other experiment parameters, we follow the default parameters of the canonical FPN [40]. In the evaluation, we employ the standard COCO metrics average precision (AP) metrics: mAP (averaged AP over IoU thresholds from 0.5 to 0.95), $AP_{@.5}$, $AP_{@.75}$, $AP_s$ (area$\in$(0,$32^2$]), $AP_m$ (area$\in$($32^2$,$96^2$]), and $AP_l$ (area$\in$($96^2$,$\infty$)).

## 4.2 Performance Analysis of DoubleFPN

Table 3 shows the evaluation result on the test set of the $M^2$SODAI dataset. As a baseline detector, we use a widely used uni-modal object detector, Faster R-CNN [40] for all benchmark results[5]. For comparison, we add an early fusion method with simple convolution layers and late fusion methods modified from DetFusion [44] and UA-CMDet [43].

**Comparison with uni-modal object detection** We first compare the DoubleFPN method and the uni-modal methods, which use either RGB or HSI data. First, we can see that the DoubleFPN method outperforms all other uni-modal methods in most of the metrics. This means that the DoubleFPN method significantly reduces the number of false positive bounding boxes by using the HSI data as a complement to RGB data. Second, when HSI data is used as a substitute for RGB data, the performance of object detection is significantly lower than that of the methods using only RGB data. This is because HSIs have relatively lower resolution than RGB images, so it is difficult to infer accurate shapes of bounding boxes even if they know whether the target objects exist or not. As a result, the benchmark results in Tab. 3 show that HSIs are suitable as a complement to RGB images, but are not yet sufficient enough as a substitute.

---

[4]The largest batch size in our GPU configuration.

[5]Although we have tried to train with recent object detectors such as TOOD [10] (mAP of 38.8 % with ResNet-50 backbone) and VFNet [55] (mAP 40.9 % of with ResNet-50 backbone), Faster R-CNN (mAP of 44.4 %) performs better under the training from the scratch settings. We note that for enhanced performance with HSI data, there's a need for either a representative pre-trained backbone layer or an improved training method for the latest detectors from the ground up.

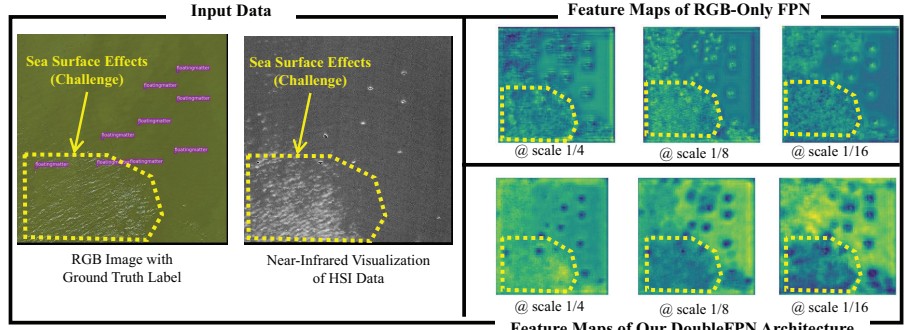

Figure 7: Feature maps with sea surface effects. The strongest feature among channels is selected for each pixel for visualization.

**Comparison with multi-modal methods**    In Tab. 3, the DoubleFPN method outperforms the early fusion method, where the reason would be that our method uses the HSI feature maps as attention maps on RGB feature maps, whereas the early fusion method directly fuses the feature maps in the backbone layer. More specifically, compared to the late fusion (UA-CMDet and DetFusion) methods, our method has a higher mAP, since late fusion methods cannot jointly fuse the feature maps of RGB and HSI data.

**Comparison with and with-out sea surface effects**  Table 4 shows the detection performance on the data with sea surface effects. By comparing Tabs. 3 and 4, the AP metrics of multi-modal methods are steady regardless of the sea surface effects; however, the AP metrics

Table 4: Benchmark result for data with sea surface effects.

| neck layer | RGB/HSI | mAP | AP$_{@.5}$ | AP$_{@.75}$ | Ship | Float. Mat. |
|---|---|---|---|---|---|---|
| **DoubleFPN (ours)** | ✓/✓ | **42.2** (↓ 2.2) | 82.3 | 31.2 | 41.7 | **42.8** |
| FPN (RGB) | ✓/✗ | 35.1 (↓ 4.4) | 73.9 | 30.0 | 42.2 | 28.0 |
| UA-CMDet | ✓/✓ | 38.9 (↓ 3.0) | 82.2 | 33.4 | 43.4 | 34.4 |
| DetFusion | ✓/✓ | 39.7 (↓ 2.3) | **83.3** | 30.2 | **43.5** | 35.9 |
| Early fusion | ✓ / ✓ | 40.4 (↓ 2.5) | 79.4 | **34.4** | 42.0 | 38.9 |

*Best: **bold and underline**, second-best: underline.
**(): mAP differences for the overall sea data results.

of the uni-modal methods have more degradation with many false positive bounding boxes if there are sea surface effects. This shows that multi-modal detection can perform more robust object detection for maritime object detection by leveraging the HSI data. For visualization, in Fig. 6, we show some samples of the object detection results and ground truth annotations on the data with sea surface effects. From the figure, we can see that the multi-modal methods propose more accurate bounding box estimations. For more examples of the benchmark results, please refer to Appendix G.

### 4.3  Visualization Analysis

Figure 7 visualizes feature maps of the DoubleFPN and the RGB-only canonical FPN. As depicted in the figure, the input data have strong sea surface effects in the bottom-left corner, which are the challenge. To the right of the input data image, we depict the feature maps of the RGB-only FPN, which are vulnerable to the sea surface effects. For example, the feature maps of the RGB-only FPN are not clear. On the other hand, in the lower part, the feature maps of the DoubleFPN are drawn, where the DoubleFPN fuses the RGB and HSI backbone outputs in order from low resolution to high resolution. As RGB and HSI data are fused, the feature maps of the DoubleFPN become clearer. Therefore, DoubleFPN can estimate the bounding boxes more accurately than RGB-only FPN method by delivering clearer feature maps to the detector.

## 5  Discussion

**Summary**    Our work addresses the problem of maritime object detection in aerial images using two types of data: RGB and HSI. To this end, we created the M$^2$SODAI dataset, which is the first dataset composed of bounding box annotations, RGB, and HSI data. We propose a multi-modal object detection framework that fuses high-resolution RGB and low-resolution HSI data. Our extensive experiments confirm the robustness of our object detection model on maritime object detection.

**Limitations** The limitations of our work are three-fold. 1) There is room for performance enhancement by having pre-trained backbone networks HSI data and multi-modal detectors instead of Faster R-CNN. 2) When we collect the data, the weather is always sunny. A future research direction is to enhance the object detection performance by proposing a new neural network architecture or to collect data in various weather conditions (*e.g.* foggy, rainy, etc.) or main/sub categories (*e.g.* buoys, rescue boats, cars, buildings, etc.). Hopefully, the atmospheric correction, typically applied during HSI data collection, can adjust for unwanted weather conditions to simulate sunny conditions, thereby allowing our data to serve as a more general representation[15]. Additionally, as the dataset has been gathered in South Korea, there may exist potential biases in the data, such as variations in the object's distribution, the condition of the oceans, and the types of ships that are commonly used in the region. 3) The data collection scenario presented in this paper requires actual aircraft and expensive HSI sensors, resulting in significant financial costs. We believe that this paper will inspire relatively low-cost drone-based data collection methods and maritime surveillance systems with HSI data. 4) Techniques for image fusion that incorporate extra HSI demonstrate greater delays in comparison to those methods that rely solely on RGB. As a result, we have advocated for forthcoming research into a 3D CNN-based feature mapping for HSI, emphasizing an approach that is both more computationally streamlined and adept at extracting essential features.

**Societal impact and ethics consideration** First, the $M^2$SODAI dataset offers a new perspective on maritime object detection, which can bring about positive societal effects in various applications such as maritime safety and national defense. A typical negative societal impact during aerial data collection is capturing sensitive areas, such as military zones or private areas. We have carefully reviewed this aspect, and we ensure that our flight areas are limited to non-military zones and non-private areas as shown in Fig. B.1.

**Usefulness of $M^2$SODAI** In our benchmark, $M^2$SODAI has demonstrated the ability to enhance object detection accuracy using HSI data to complement existing high-resolution optical images. We have strong confidence that this dataset will not be limited to object detection tasks but can also be sufficiently utilized for other tasks, such as RGB to HSI reconstruction tasks.

## Acknowledgment

This research was supported by a grant from Endowment Project of "Development of Open Platform Technologies for Smart Maritime Safety and Industries" funded by Korea Research Institute of Ships and Ocean engineering (PES4880), and Korea Institute of Marine Science and Technology Promotion (KIMST) funded by the Ministry of Oceans and Fisheries (RS-2022-KS221606).

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
