# OpenReview forum: "M$^{2}$SODAI: Multi-Modal Maritime Object Detection Dataset With RGB and Hyperspectral Image Sensors"
_NeurIPS.cc/2023/Track/Datasets_and_Benchmarks — NeurIPS 2023 Datasets and Benchmarks Poster_

### Official Review · Reviewer_V53X · 2023-07-17
**The comments and suggestions on M2SODAI dataset paper.**

**Rating:** 4
**Confidence:** 4
**Clarity:** No.

**Strengths:**

The dataset uses hyperspectral data to solve the false alarm problem caused by sea surface effects in RGB images, promoting the development of multi-modal data in maritime object detection, which has good application prospects in surveillance, monitoring, and active rescuing.

**Additional Feedback:**

No.

**Correctness:**

Yes, but the benchmark is not adequate. The evaluation methods are proposed several years ago, which are not novel and advanced enough. The experiment design is also unfair.

**Documentation:**

Yes.

**Ethics:**

No.

**Limitations:**

The authors discussed the limitations of this dataset in terms of backbone network and diversity of weather. If there’s potential negative societal impact, we also suggest the authors to complement them.

**Opportunities For Improvement:**

1. The contributions are not outstanding enough. Data collection and data preprocessing are not innovative and not suitable as contribution. Benchmark can indeed be one of the contributions, but it should mainly introduce the comprehensiveness of the experiments and analysis, rather than introducing new methods first. Perhaps the contribution can also be summarized in terms of dataset size, the categories of target and scenario categories.
2. The chapter arrangement is not reasonable. The related work is the only subsection of section 1, which does not form the logical structure from whole to branch.
3. Section 2 should begin with an overview of the dataset construction process. The word “method” in the first subtitle is not appropriate.
4. The motivation should be discussed in section 1. When I read your paper, I couldn’t understand the significance of the hyperspectral images until the end of page 5. You should explain the challenge and motivation clearly in section 1.
5. Figure 4 can also be placed in section 1 as an argument to support your motivation.
6. The benchmark is inadequate. The methods in your benchmark are proposed several years ago, which are not novel and advanced enough. Add some methods proposed in recent three years in your benchmark.
7. The experiment of multi-modal method is unreasonable. The comparison can only prove the advantage of feature-level fusion over early fusion and late fusion, as there’s no feature-level fusion method for comparison in your experiment.
8. There are many advanced fusion methods in pixel-level and decision-level, but the multi-modal methods in your experiment are too simple to represent the early fusion and late fusion methods.

**Relation To Prior Work:**

Yes.

**Summary And Contributions:**

This paper presents a RGB-hyperspectral multi-modal dataset for maritime object detection, which includes 1257 synchronized image pairs and 11892 instances. Compared with the previous work, the amount of data in this paper has been improved. Considering its value in maritime application scenarios, the presented study is also interesting. Hyperspectral images solve the false alarm problem caused by sea surface effect in RGB images, and RGB images also complement the spatial details of hyperspectral images.

---

> ### Author Response · Authors · 2023-08-20
> **Response to  Author Reviewer V53X (Part 1/2)**
>
> Thank you for these constructive reviews that help us improve the quality of the manuscript.
> The point-by-point responses are written as follows:
>
> **1. [Contribution]**
>
> > Comment: The contributions are not outstanding enough. Data collection and data preprocessing are not innovative and not suitable as contribution. Benchmark can indeed be one of the contributions, but it should mainly introduce the comprehensiveness of the experiments and analysis, rather than introducing new methods first. Perhaps the contribution can also be summarized in terms of dataset size, the categories of target and scenario categories.
>
> Thank you for your sharp comments.
> The authors agree that benchmarks should be proposed more clearly before presenting new methods. I also agree that contributions to the nature of the dataset (dataset size, etc.) should be emphasized.
> The data collection process is financially and time-challenging; thus, we believe that it can be a sub-contribution to the data collection. Also, since we added a contrast enhancer block in the image registration, it can be a sub-contribution to the data collection.
>
> - To cope with the reviewer’s comment we carefully revised the contributions in Lines 56-67.
>
> **2. [Rearranging on Related work section]**
>
> > Comment: The chapter arrangement is not reasonable. The related work is the only subsection of section 1, which does not form the logical structure from whole to branch.
>
> We agree with the reviewer that the related work section does not form any logical structure form; thus, we moved the related work section to the appendix and make it as a independent section.
>
> **3. [Clearness of the representations]**
>
> > Comment: Section 2 should begin with an overview of the dataset construction process. The word “method” in the first subtitle is not appropriate.
>
> To follow the reviewer’s suggestions, we make the related work section as a separate section and move to the appendix. Also, the expression “over view of the proposed method” is replaced by “overview of the dataset construction process”
>
> **4. [Motivation]**
>
> > Comment: The motivation should be discussed in section 1. When I read your paper, I couldn’t understand the significance of the hyperspectral images until the end of page 5. You should explain the challenge and motivation clearly in section 1.
>
> Thank you for your considerate comment. We also agree with that the motivation should be moved to Section 1. Also, to explain the motivation, we place Figure 4 in Section 1 as an argument to support your motivation.
>
> - Motivation and challenges are added in Section 1(Lines 33-38)

---

> ### Author Response · Authors · 2023-08-20
> **Response to Author Reviewer V53X (Part 2/2)**
>
> **5. [Advancements in the benchmark (Table 3, Table 4, and most of the parts in Section 4)]**
>
> > Comment: The benchmark is inadequate. The methods in your benchmark are proposed several years ago, which are not novel and advanced enough. Add some methods proposed in recent three years in your benchmark.
>
> > Comment: The experiment of multi-modal method is unreasonable. The comparison can only prove the advantage of feature-level fusion over early fusion and late fusion, as there’s no feature-level fusion method for comparison in your experiment.
>
> > Comment: There are many advanced fusion methods in pixel-level and decision-level, but the multi-modal methods in your experiment are too simple to represent the early fusion and late fusion methods.
>
> **[recent object detectors]:** We strongly agree with the reviewer’s comment that the method used in our benchmark is quite old. Hence, we have tried to add recent detectors (VfNet and TOOD); however, both of the detectors have lower performance than Faster R-CNN in `training from scratch’ setting.
>
> **[Additional fusion methods for comparison]:** To follow the suggestions, we have simulated UA-CMDet [r1] and DetFusion [r2], which are modified for more appropriate in our exper iment.  We note that both methods are kinds of late (decision-level) fusion. Also, for a pixel-level feature fusion, we added some convolution layers in front of the backbone layer inspired from UA-CMDet. As discussed in Table 3, the DoubleFPN outperforms the existing early fusion method and late fusion methods (UA-CMDet, DetFusion)
>
> **[Advanced backbone layer]:** As far as the authors can do (up to the limits allowed by the GPU RAM), ResNeXt50 is able to operate, so we newly added the experimental results for the backbone.
>
> [r1] Yiming Sun, Bing Cao, Pengfei Zhu, and Qinghua Hu. Drone-based rgb-infrared cross-modality vehicle detection via uncertainty-aware learning. IEEE Transactions on Circuits and Systems for Video Technology,  32(10):6700–6713, 2022. doi: 10.1109/TCSVT.2022.3168279.
>
> [r2] Yiming Sun, Bing Cao, Pengfei Zhu, and Qinghua Hu. Detfusion: A detection-driven infrared and visible image fusion network. In Proceedings of the ACM International Conference on Multimedia, pages 4003–4011, 2022

---

> ### Comment · Reviewer_V53X · 2023-08-29
> **Response to the revised manuscript**
>
> I have received your reply. Most of my concerns have been competently addressed, so that I will upgrade my rating. But as far as I am concerned, the article still has the following problems, so I can not give a higher score.
>
> 1. Comparison methods are still not abundant enough. The benchmark your built needs to test a large number of advanced methods and comprehensive experiment, so that this benchmark could benefit others more in the future. I'd like to see that you have tried recent detectors, but you still need to list them and discuss.
>
> 2. I think the authors still has something to improve in writing. For example, contribution is your characteristic, your innovation, what sets you apart from everyone else. It should be fascinating enough. There's no need to show your workload in data collection here because other research also paid much when constructing a large dataset.

---

> > ### Author Response · Authors · 2023-08-30
> >
> > Dear Reviewer V53X,
> >
> > Firstly, thank you for kindly pointing out the shortcomings in our response. We have made several revisions to the manuscript to address the comments. Since we specified line numbers during discussions with other reviewers, may we upload this revised version after receiving responses from all the reviewers to avoid any confusion? Below, we detail the changes made or present them verbatim.
> >
> > 1. **[Recent Detectors]** Thank you sincerely for your comments regarding our experiments on recent detectors. We'd like to kindly remind that training from scratch without a pre-trained network is a research topic in the field of object detection [r1, r2]. The Faster R-CNN has existing research and guidance on training from scratch, which we followed and subsequently produced results. However, the latest detectors lack research outcomes for scratch training, and due to the absence of clear guidelines, the training experiments took a considerable amount of time (even for RGB-only experiments). Below are the results from experiments conducted with the ResNet-50 backbone network.
> >
> > | Detector     | Neck      | mAP  | AP_{@.5} | AP_{@.75} | Ship | Float. Mat. | AP_{s} | AP_{m} | AP_{l} |
> > |--------------|-----------|------|----------|-----------|------|-------------|--------|--------|--------|
> > | Faster R-CNN | DoubleFPN | 44.4 | 84.8     | 39.3      | 55.7 | 33.1        | 35.2   | 41.7   | 61.4   |
> > | TOOD         | DoubleFPN | 38.8 | 77.3     | 34.4      | 47.5 | 30.2        | 15.3   | 36.2   | 51.4   |
> > | VFNet        | DoubleFPN | 40.9 | 79.8     | 35.9      | 52.6 | 29.2        | 17.2   | 39.5   | 57.8   |
> >
> > We have made our utmost effort for the additional experiments. As much as we'd like to include these results in the table, we are concerned that adding them might cause further confusion for the readers. Instead, we have added the results as **footnote 5 on page 5** in the main text and mentioned the need for research on both training from scratch and pre-trained backbones as follows.
> >
> > > Although we have tried to train with recent object detectors such as TOOD [10] (mAP of 38.8 % with ResNet-50 backbone) and VFNet [55] (mAP 40.9 % of with ResNet-50 backbone), Faster R-CNN (mAP of 44.4 %) performs the best under the training from the scratch settings. We note that a representative pre-trained backbone layer for HSI data or a better method for training the recent detectors from scratch is required for better performance.
> >
> > We hope this addresses the reviewer's concerns adequately.
> >
> > [r1] Hong, Weixiang, et al. "Training object detectors from scratch: An empirical study in the era of vision transformer." Proceedings of the IEEE/CVF Conference on Computer Vision and Pattern Recognition. 2022.
> > [r2] Li, Yang, Hong Zhang, and Yu Zhang. "Rethinking training from scratch for object detection." arXiv preprint arXiv:2106.03112 2021.
> >
> >
> > 2. **[Paper's contribution**] Thank you for your meticulous feedback on the contribution section. We believe the reviewer commented on the data collection part and raw data processing of our contribution. I sincerely agree that as a reviewer for the NeurIPS dataset track, the workload is a given. Hence, we also **do not consider our data collection (59 flights and 12 flight measurement campaigns...) as our unique contribution**. Accordingly, the **data collection part has been excluded** from our contributions. On the other hand, for raw data processing, we enhanced the existing method used for image matching by adding a contrast enhancer. This enhancement facilitates better alignment and is an essential element in creating a synchronized dataset, making it a unique contribution compared to other dataset constructions. Beyond this, we pondered other unique contributions, but regretfully, we could not identify additional ones except the already mentioned "first multi-modal, bounding-box-labeled and synchronized dataset with 11,892 instances and 1,257 images" in our paper. We genuinely regret this aspect. However, we believe we have done our best.
> >
> > Lastly, we deeply appreciate the revised rating.
> > We sincerely hope this response meets the reviewer's concern, regardless of the further score change.
> >
> > Thank you.
> >
> > Authors of M2SODAI Dataset.

---

### Official Review · Reviewer_WDCT · 2023-07-21
**Interesting multimodal data for maritime object detection. Consent acquisition, and the benefits of combining RGB and HSI data (400nm to 700nm range) for maritime applications are required.**

**Rating:** 7
**Confidence:** 3

**Strengths:**

Significance of the Contribution:
The new dataset for maritime object detection is significant to the research focused on detecting objects on or near the sea surface. The inclusion of HSI sensor is useful to differentiate between the sea surface and objects floating on the water.

Relevance to the Broader Research Community:
The maritime object detection dataset appeals to researchers in computer vision, remote sensing, and marine sciences, encouraging advancements in multimodal data fusion for maritime object detection research.

Quality of the Research:
The paper demonstrates a commendable level of detail in the dataset creation process, encompassing data collection, annotation, and evaluation.

Ethical and Social Implications:
The potential applications of the dataset, such as enhancing surveillance and rescue operations, can have positive impacts on the maritime industry and society as a whole.

**Additional Feedback:**

I want to thank the authors for their effort in collecting this multimodal dataset combining RGB and HSI data for maritime object detection. However, I have a question regarding the rationale for combining RGB and HSI data, considering that the RGB information is already contained implicitly within the HSI data. Could you please clarify the usefulness of including RGB data for maritime object detection?

Also, regarding the spectral range of the HSI data, if the spectral range spans from 400nm to 1000nm, I would like to understand whether it is sufficient for maritime applications, given that the environment mainly comprises water components.

**Clarity:**

The paper is well-written, with proper references and a comprehensive checklist.





**Correctness:**

The dataset construction process appears to be thorough, with detailed information about data collection, data pair alignment, bounding box annotation, and evaluation with state-of-the-art methods. However, a notable concern arises from the authors' claim of obtaining consent (4.(d) checklist) without further discussion in the paper regarding how this consent was actually obtained, leaving this aspect unexplained.

**Documentation:**

The data collection process, including aligning image pairs and bounding box annotation, is well-documented. A subset of the dataset is accessible via a provided URL. However, the paper lacks information on the full dataset's public availability timeline and duration.

**Ethics:**

The authors claimed full compliance with the ethics review guidelines according to the checklist. However, there is no supporting document available to verify this claim.

**Limitations:**

The authors acknowledged limitations in Section 5, but lacked explicit discussions on potential societal impacts and biases in data annotation.

A few suggestions:
- Address potential privacy concerns in maritime surveillance using object detection algorithms.
- Explore biases in data annotation and their impact on detection accuracy.
- Discuss approach limitations (e.g., specialized equipment, cost) for real-world implementation.

**Opportunities For Improvement:**

Significance of the Contribution:
To reinforce the significance of the contribution, the authors should provide additional details on the dataset's adoption and demonstrate its utility in addressing real-world challenges.

Relevance to the Broader Research Community:
While the paper focuses on object detection for maritime applications, the authors can discuss potential cross-domain relevance to other areas of computer vision and remote sensing, so that to increase its appeal to researchers beyond the maritime domain.

Quality of the Research:
Despite the dataset being collected during sunny days, it is essential for the authors to thoroughly discuss how well it captures real-world maritime settings and scenarios to ensure its representativeness.  Also, the authors need to address potential biases in data collection and annotation to enhance the dataset's credibility and applicability in practical situations.

Ethical and Social Implications:
The paper should include more explicit discussions addressing privacy and data usage considerations, particularly if the dataset is intended for maritime surveillance applications.




**Relation To Prior Work:**

The paper differs from prior work by fusing RGB and HSI data for maritime object detection. This approach facilitates the detection of both large and small vessels, effectively addressing limitations seen in earlier methods.

**Summary And Contributions:**

The paper introduces a new dataset called "Multi-modal ship and floating matter detection in Aerial Images (M2SODAI)" specifically designed for object detection in maritime settings. The dataset contains 11,892 instances of 23 object categories, and it provides synchronized image pairs of RGB and hyperspectral data. Each instance in the M2SODAI dataset comes with bounding box labels, facilitating a more comprehensive evaluation and in-depth analysis of object detection performance in both RGB and hyperspectral modalities.

---

> ### Author Response · Authors · 2023-08-20
> **Response to Author Reviewer WDCT**
>
> Thank you for your kind suggestions. We believe that the reviewer's suggestion highly help enhance the quality of the manuscript, mainly for the limitation.
>
> **1. [Significance of the Contribution]**
>
> > Comment: To reinforce the significance of the contribution, the authors should provide additional details on the dataset's adoption and demonstrate its utility in addressing real-world challenges.
>
> New additions are added to explain the real-world challenge (sun glint rejection) as an example of sea surface removal and justification for data adoption. (in Lines 33-38)
>
> **2. [Relevance to the Broader Research Community]**
>
> > Comment: While the paper focuses on object detection for maritime applications, the authors can discuss potential cross-domain relevance to other areas of computer vision and remote sensing, so that to increase its appeal to researchers beyond the maritime domain.
>
> The synchronized dataset we presented can be used not only for simple object detection but also for RGB to HSI image synthesis, which can be used to expand existing aerial data. Also, it seems that HSI images can be super-resolution by combining RGB images and HSI images.
>
> - To represent the response in the manuscript, we newly added Lines 279-282.
>
> **3. [Quality of the Research]**
>
> > Comment: Despite the dataset being collected during sunny days, it is essential for the authors to thoroughly discuss how well it captures real-world maritime settings and scenarios to ensure its representativeness.
>
> We collected aerial data for clear weather as the reviewer stated. Fortunately, the atmospheric correction, usually implemented in HSI data collection, can transform unfavorable weather conditions into the appearance of sunny weather, thus rendering our data more universally applicable.
>
> - To comply with the review, we newly added a reference [r1] and Lines 263-265.
>
> [r1] Gao, Bo-Cai, et al. "Atmospheric correction algorithms for hyperspectral remote sensing data of land and ocean." Remote sensing of environment 113 (2009): S17-S24. .
>
> **4. [Potential bias]**
>
> > Comment: Also, the authors need to address potential biases in data collection and annotation to enhance the dataset's credibility and applicability in practical situations.
>
> Even though we collected M2SODAI dataset on 11 different spots,, since the data collection took place in South Korea, certain inherent biases might be present in the dataset. These biases could relate to the distribution of objects, the condition of the oceans, or the types of ships that are predominantly used within the region.
>
> - We newly added the potential biases in Lines 263-265.
>
> **5. [Ethical and Social Implications]**
>
> > Comment: The paper should include more explicit discussions addressing privacy and data usage considerations, particularly if the dataset is intended for maritime surveillance applications.
>
> We sincerely thank the author for his meticulous consideration. A supporting document was included in the supplementary stating that the authors considered ethics review regarding privacy and data usage.
>
> - We newly added a paragraph showing the ethical and social implications in Lines 275-280.
>
> **6. [Additional Limitations]**
>
> > Comment: Discuss approach limitations (e.g., specialized equipment, cost) for real-world implementation.
>
> We have added information stating that the collection of the dataset requires actual aircraft, and due to the need for expensive RGB and HSI sensors, there is a distinct financial limitation.
>
> - To reflect the reviewer’s comment, we added a costly limitation in Lines 268-270.

---

### Official Review · Reviewer_jUWJ · 2023-07-21
**Review for M2SODAI**

**Rating:** 8
**Confidence:** 5

**Strengths:**

The biggest strength of this paper is the collection of a novel RGB + HSI imagery. The authors collect a substantive new dataset via low-altitude airplane, coregister and annotate the imagery, and then split it into configurations for a variety of object detection tests.

A secondary strength of this paper is the novel training strategy and model architecture that the authors propose with their DoubleFPN. This new architecture + approach is theoretically substantiated and well tested, and compared to other existing approaches that would be familiar to any reader.

The general presentation of the paper is also high-quality. Figures and tables are clean, informative, and well-deployed throughout the paper.

**Additional Feedback:**

N/A. Well done!

**Clarity:**

The paper is well written. There are a couple awkwardly worded sentences throughout the paper, but nothing material or anything that takes away from the paper.

**Correctness:**

Yes, the submissions seem generally correct. The dataset and evaluation methods are constructed in a sound way.

**Documentation:**

Yes.

**Ethics:**

Not applicable.

**Limitations:**

I cannot see an potential negative societal impact of this work.

**Opportunities For Improvement:**

There are some issues I would like to see addressed during this paper's revision:

- First, the authors should provide a physics-based, theoretical justification for why the HSI imagery improves object detection performance when used in concert with RGB imagery. The HSI imagery here is collected between wavelengths of 400 and 1000nm, which is entirely covered by the visual spectrum + the early part of the NIR spectrum. Water is highly absorptive in the SWIR portion of the EM spectrum, but those wavelengths are not presented here; as water is less absorptive of visible + NIR waves, it is theoretically more difficult to differentiate water from other reflective, non-water objects using only these spectra. What is it about the 400nm-1000nm spectrum captured by the HSI instrument that lead to better performance over the RGB imagery already covering 400nm-700nm?

- I would like to see the authors adjust their experimental approach. The results presented in Tables 3 and 5 show a clear advantage to the proposed DoubleFPN approach compared to other single modality baselines. However, it is unclear how much of this improvement is due to 1) the proposed model + training strategy, or 2) the fact that this approach uses both types of imagery compared to all others that use only 1. The authors should include single FPN experiments for both the RGB and HSI imagery separately so this effect can be teased out.

- Relatedly, I am a bit confused by the fact that the proposed approach is presented as "Ours + Faster R-CNN". How does the faster RCNN factor in here? It seems like the double FPN network training strategy is distinct from the Faster RCNN one.

- The authors should experiment with a more SOTA single-stage detection model, such as Yolo-v8. Yolov8 has shown significant improvements over Yolov3 and even two-stage detectors.

- The authors should add more detail about the labeling efforts in the main text. For instance, exactly what classes of objects are labeled (only ships and sea surface effects, or anything more)?  Also what tool is used for the labeling, how many people collected the labels, what instructions were they given, etc.

- The authors should clarify whether the image coregistration strategy detailed in lines 122-154 is a standard approach or something novel.

- The authors should specify in the abstract and/or introduction what the spatial resolution of the imagery is (i.e. ahead of table 2).

- The authors should also discuss how CNNs and deep learning strategies have been adapted to HSI imagery or other related tasks where the computer vision modeling input contains 10s-100s of channels.

- In the implementation section, the authors should specify whether the hyperparameters selected are the result of experimentation or are just values that the authors picked and found worked well enough (this is fine either way).

- There is a typo in line 283 -- "Analaysis"

**Relation To Prior Work:**

Yes, the introduction and literature review are sufficient.

**Summary And Contributions:**

The M2SODAI paper makes a number of clear contributions to the literature. First, it introduces a novel dataset for maritime vessel detection that contains both RGB and HSI imagery. Second, it experiments with a number of deep learning training strategies to compare detection methods, and introduces a novel CNN architecture that leverages a double feature pyramid network.

Overall, this is a strong paper and deserves acceptance. There are a few minor issues with the paper that I would like to see addressed that I describe below.

---

> ### Author Response · Authors · 2023-08-20
> **Response to Reviewer jUWJ (Part 1/2)**
>
> Sorry for our lateness. We have revised the manuscript to satisfy the comments the reviewer raised. We hope our point-wise responses make satisfactory.
>
> **1. [Justification for HSI imagery]**
>
> > Comment: First, the authors should provide a physics-based, theoretical justification for why the HSI imagery improves object detection performance when used in concert with RGB imagery. The HSI imagery here is collected between wavelengths of 400 and 1000nm, which is entirely covered by the visual spectrum + the early part of the NIR spectrum. Water is highly absorptive in the SWIR portion of the EM spectrum, but those wavelengths are not presented here; as water is less absorptive of visible + NIR waves, it is theoretically more difficult to differentiate water from other reflective, non-water objects using only these spectra. What is it about the 400nm-1000nm spectrum captured by the HSI instrument that lead to better performance over the RGB imagery already covering 400nm-700nm? I would like to see the authors adjust their experimental approach.
>
> As the reviewer has noted, water exhibits a characteristic where its reflectance decreases as the wavelength increases, especially near the SWIR band. Water tends to have a lower level of reflectance, making it easily distinguishable from other objects. However, even utilizing the wavelengths in the NIR region, water exhibits a pattern of sharply reduced reflectance between 700nm-900nm, as seen in Figure 1 and [r1], distinguishing it from other objects. Therefore, our HSI can contribute to distinguishing objects from sea surface effects, like rejecting the sub-glints.
>
> - To comply with the comment, we added justification of HSI imagery in Lines 33-38.
>
> [r1] P. Duan, J. Kang, X. Kang, P. Ghamisi and S. Li, "Sun Glint Removal of Hyperspectral Images via Texture-Aware Total Variation," IGARSS 2020 - 2020 IEEE International Geoscience and Remote Sensing Symposium, Waikoloa, HI, USA, 2020, pp. 7005-7008, doi: 10.1109/IGARSS39084.2020.9323871.
>
> **2. [Clearness of the results]**
>
> > Comment: The results presented in Tables 3 and 5 show a clear advantage to the proposed DoubleFPN approach compared to other single modality baselines. However, it is unclear how much of this improvement is due to 1) the proposed model + training strategy, or 2) the fact that this approach uses both types of imagery compared to all others that use only 1. The authors should include single FPN experiments for both the RGB and HSI imagery separately so this effect can be teased out. Relatedly, I am a bit confused by the fact that the proposed approach is presented as "Ours + Faster R-CNN". How does the faster RCNN factor in here? It seems like the double FPN network training strategy is distinct from the Faster RCNN one.
>
> Thank you for your comment on the paper’s clearness. We sincerely apologize for the confusion. Faster R-CNN was merely used as a detector, which seems to have caused the misunderstanding. The DoubleFPN is a distinct one with the detector (Faster R-CNN). To effectively present the benefits of using both types of data, we have altered the way they are displayed in the tables and made a clear comparison with a single FPN." That is, to comply with the comment, we removed all the detectors without Faster R-CNN-based ones.
>
> - We modify the Tables 3, 4 and G.1 to comply with the comment.
>
> **3. [Further research with advanced detectors]**
>
> > Comment: The authors should experiment with a more SOTA single-stage detection model, such as Yolo-v8. Yolov8 has shown significant improvements over Yolov3 and even two-stage detectors.
>
> We experimented with advanced detector algorithms (such as VfNet and TOOD), which are recent object detectors like Yolo v8. However, training them from scratch did not lead to proper convergence, and their performance was even worse than Faster R-CNN. This would likely be the case with Yolo v8 as well. Since training from scratch is a research topic in the object detection domain [r1, r2], we conducted all benchmarks through Faster R-CNN, where successful learning is guaranteed. While it is regrettable that Yolo v8 was not included, the authors firmly believe that this discussion has undoubtedly enhanced the clarity and quality of the paper."
>
> [r1] [Hong, Weixiang, et al. "Training object detectors from scratch: An empirical study in the era of vision transformer." Proceedings of the IEEE/CVF Conference on Computer Vision and Pattern Recognition. 2022.]
> [r2] Li, Yang, Hong Zhang, and Yu Zhang. "Rethinking training from scratch for object detection." arXiv preprint arXiv:2106.03112 (2021).

---

> ### Author Response · Authors · 2023-08-20
> **Response to Reviewer jUWJ (Part 2/2)**
>
> **4. [Labeling policy details]**
>
> > Comment: The authors should add more detail about the labeling efforts in the main text.  For instance, exactly what classes of objects are labeled (only ships and sea surface effects, or anything more)? Also what tool is used for the labeling, how many people collected the labels, what instructions were they given, etc.
>
> Thank you for this recommendation. In the main manuscript, we newly added the following instructions:
> - We note that the following instances are labeled as floating matters: buoys, rescue tubes, small lifeboats, surfboards, and humans (mannequins) with life vests. Also, for the ship class, we annotated bounding boxes on steamboats, cruise ships, fishing boats, sailboats, rafts, and other ship categories. For labeling, two of the authors annotated target instances by using Labelme [46], in which the minimum size box containing each object was set as the policy, and multiple checks were performed.
>     - This discussion is reflected in  Lines 121-136.
>
> **5. [spatial resolution before Table 2]
>
> > Comment: The authors should specify in the abstract and/or introduction what the spatial resolution of the imagery is (i.e. ahead of table 2).
>
>
> Thank you for your careful reading. In the interest of detail and clarity, the authors have willingly specified the spatial resolutions in the introduction.
>
> **6. [image coregistration strategy]**
>
> > Comment: The authors should clarify whether the image coregistration strategy detailed in lines 122-154 is a standard approach or something novel.
>
> The gray-scale-based image registration is well known work; however, we found that the method doesn’t work for our dataset. For more accurate image registration, we added a contrast enhancer to the RGB/HSI gray-scaled images, which is our contribution.
>
> - To comply with the comment we have revised Lines 65-67.
>
> **7. [CV modeling for 10s-100s channels]**
>
> > Comment: I would like to see the authors adjust their experimental approach.  The authors should also discuss how CNNs and deep learning strategies have been adapted to HSI imagery or other related tasks where the computer vision modeling input contains 10s-100s of channels.
>
> If all 127 channels were to be utilized, options such as 3D CNN might be employed in computer vision. However, we used PCA to extract the principal components, and during this process, even though the spectral features of the data were retained, much of the spectral correlation was lost. Therefore, we utilized CNN, which is among the commonly used methods, and found no significant performance difference when compared with 3D CNN. But, the authors believe that there is room for enhancing the results by using 3D CNN.
>
> - Hence, we added further research directions on efficient data  backbone for the HSI data (Lines 272-274.)
>
> **8. [Clearness of the Hyperparameter Selection]**
>
> > Comment: In the implementation section, the authors should specify whether the hyperparameters selected are the result of experimentation or are just values that the authors picked and found worked well enough (this is fine either way).
>
> Our hyperparameters were adopted from the baseline learning strategy of Faster R-CNN with FPN (from scratch). To comply with the comment, we added a new instruction in Lines 683-684.
>
> **9. [Correction Typo]**
>
> > Comment: There is a typo in line 283 -- "Analaysis"
>
> We sincerely thank you for the correction. I corrected the typo.

---

### Official Review · Reviewer_cscm · 2023-07-21
**The suggested dataset, M^2 SODAI, is unique in that it is the first i) bounding-box-annotated, ii) synchronized multi-modal, and iii) aerial RGB and HSI data. The new network, DoubleFPN, can perform better in multi-modal scenarios.**

**Rating:** 7
**Confidence:** 3
**Clarity:** Yes. The paper is mostly well written.

**Strengths:**

1.	Multi-modal Approach: The paper introduces a multi-modal approach by combining RGB and HSI data for object detection. This approach outperforms uni-modal methods that use either RGB or HSI data alone. The DoubleFPN method significantly reduces the number of false positive bounding boxes by leveraging the complementary information provided by HSI data.
2.	Robustness to Sea Surface Effects: The paper addresses the challenge of robust object detection in the presence of sea surface effects. Through data analysis, the authors demonstrate that HSI data can effectively distinguish target objects from the background, even with low resolution. This robustness is crucial for accurate object detection in aerial images.
3.	Performance Improvement: The proposed DoubleFPN method outperforms other multi-modal methods, such as early fusion and late fusion, in terms of mean average precision (mAP). It achieves higher mAP compared to uni-modal methods that use only RGB data. This improvement in performance highlights the effectiveness of the multi-modal approach.
4.	Dataset Availability: The paper introduces the M^2 SODAI dataset, which includes synchronized image pairs of RGB and HSI data. This dataset provides a valuable resource for researchers working on object detection in aerial images.
5.	Visualization and Analysis: The paper includes visualizations and analysis of feature maps, demonstrating the effectiveness of the DoubleFPN method in handling sea surface effects. These visualizations provide insights into the performance of the proposed method and support the claims made in the paper.
6.	The paper is mostly well-written.
7.	The authors provide a project page which contain codes, dataset, and visual materials.


**Additional Feedback:**

Q) Do you reckon a model trained with M^2 SODAI would perform better than any model trained with other aerial maritime datasets?

**Correctness:**

Yes. The submission is made correctly, and the dataset is constructed in a sound way.

**Documentation:**

Yes. The authors have provided sufficient details on data collection, organization, and availability. They have also provided a project page which includes dataset, code (not yet but they mentioned they will), paper, and other useful information of their work. How they maintain and how they use for ethical purposes have not been discussed.

**Ethics:**

No. I do not suspect any ethical concerns.

**Limitations:**

Yes. The authors have addressed the limitations in the paper as follows:

1.	Performance Enhancement: The authors mention that there is room for performance enhancement by modifying state-of-the-art backbone networks for feature fusion, such as the vision transformer (ViT). This suggests that further improvements can be made to the proposed multi-modal object detection framework by exploring different network architectures.

2.	Limited Weather Conditions: The data collection for this study was conducted under sunny weather conditions. As a result, the dataset does not include variations in weather conditions, such as foggy or rainy conditions. This limitation suggests that the object detection performance of the proposed method may need to be further evaluated and enhanced under different weather conditions.

3.	Class number: The final bounding boxes are classified as only two classes. The authors have mentioned they will subdivide the classes (e.g., rescue boats, buoys, etc.).

Potential negative societal impacts have not been discuessed.


**Opportunities For Improvement:**

1.	There is no mention of inference speed. FPN has improved performance, memory efficiency, and speed compared to the previous pyramid methods. DoubleFPN would be slower since it is bigger in size than FPN and uses two inputs. In Section 4.1, two RTX 3090 GPUs were used, and the batch size was fixed to 1. The reason why the batch size was fixed to 1 was also not explained. It would have been better if they could compare the performance with memory efficiency and speed.
2.	Because HSI images contain channels of various wavelengths, each channel would contain intrinsic information. They acquired HSI data on 127 channels. It would have been better if object detection speed and performance had been ablated when using part of 127 channels as if they used “Near-infrared” image in section 4.3.
3.	It is questionable whether any model learned with M^2 SODAI dataset performs better than any other model learned with other dataset. Even if the latter is a model trained with a single image, it is questionable which model would perform better if the same arial maritime image is inputted. Have you considered performance comparison with models learned with other datasets?
4.	Minor error: Table 1; “dehzing” -> dehazing
5.	Minor error: 24 line;
“there have been sea surface maritime surveillance systems such as buoys and ships”
Buoys and ships are not systems. The sentence could confuse readers.


**Relation To Prior Work:**

Yes. The authors clarified the difference from previous works with tables and related works section.

**Summary And Contributions:**

The paper introduces the M^2 SODAI dataset, which consists of synchronized image pairs of RGB and HSI data for object detection in aerial images. The authors propose a new multi-modal extension of the feature pyramid network (FPN) called DoubleFPN. The motivation behind this research is to develop a robust object detection method for environmental conditions such as solar reflection or waves which they call Sea Surface Effects. The paper includes data analysis showing that HSI data can distinguish target objects from the background by identifying their intrinsic characteristics.

---

> ### Author Response · Authors · 2023-08-20
> **Respond to Reviewer cscm**
>
> Firstly, we sincerely thank you for the detailed and constructive review of our manuscript. The point-by-point responses to the review are as follows. We believe that this discussion is especially helpful in developing the motivation.
>
> **1. [Batchsize configuration]**
> > Comment: There is no mention of inference speed. FPN has improved performance, memory efficiency, and speed compared to the previous pyramid methods. DoubleFPN would be slower since it is bigger in size than FPN and uses two inputs. In Section 4.1, two RTX 3090 GPUs were used, and the batch size was fixed to 1. The reason why the batch size was fixed to 1 was also not explained.
>
> The utilization of a batch size of 1 was necessitated by the substantial GPU memory requirement of over 16GB, even when employing 16-bit floating-point numbers with automatic mixed precision (AMP). Therefore, under these conditions, only a batch size of 1 could be implemented.
>
> - We newly added footnote 4 on page 7 based to respect the comment.
>
> **2. [Memory Efficiency]**
>
> > Comment: It would have been better if they could compare the performance with memory efficiency and speed.
>
> We have carefully included this information in Table G2, located in the appendix. Image fusion methods that work with supplementary HSI demonstrate longer delays than the methods employing only RGB. Consequently, we have encouraged further research into 3D CNN-based feature mapping for HSI, with the goal of creating a more computationally effective and efficient approach to feature extraction.
>
> - We kindly added inference time and GPU Usage column in the Table G2 and lines 768-770.
> - We also added further research directions on memory-efficient feature extraction and object detection. (in lines 272-274)
>
> **3. [Ablation study on each spectral channel]**
>
> > Comment: Because HSI images contain channels of various wavelengths, each channel would contain intrinsic information. They acquired HSI data on 127 channels. It would have been better if object detection speed and performance had been ablated when using part of 127 channels as if they used “Near-infrared” image in section 4.3.
>
> As mentioned, each wavelength channel of HSI contains unique information. By comparing the object detection speed and performance when using some channels, such as near-infrared visualization as shown in Section 4.3's visualization, we can understand the influence of each channel, and we completely agree that this would have resulted in a truly perfect analysis. However, conducting related experiments takes too long due to the excessive combination of channels, and the method is intricate.
>
> - We will try to add motivations of channel-wise features to the dataset home page when the ablation study finishes.
>
> **4. [Performance comparison with models learned with other datasets]**
>
> > Comment: It is questionable whether any model learned with M^2 SODAI dataset performs better than any other model learned with other dataset. Even if the latter is a model trained with a single image, it is questionable which model would perform better if the same arial maritime image is inputted. Have you considered performance comparison with models learned with other datasets? (Almost the same one to Additional Feedback)
>
> Thank you very much for your comments. Indeed, we have not been able to compare the performance with a model trained on an RGB-only dataset. The absence of standardized labels is the primary reason for this. However, we believe that the M2SODAI has advantages, especially in identifying patterns and visually similar small objects in environments where the sea surface effect is significantly presented.
>
> **5. [Minor errors]**
> > Minor error: Table 1; “dehzing” -> dehazing
>
> > Minor error: 24 line; “there have been sea surface maritime surveillance systems such as buoys and ships” Buoys and ships are not systems. The sentence could confuse readers.
>
> We have revised the manuscript to follow up on the reviewer’s suggestions.
>
> **6. [Maintainance]**
>
> > Comment: How they maintain and how they use for ethical purposes have not been discussed.
>
> Thank you for your constructive review. We kindly added the maintenance plan in Appendix C. (Lines 564-567)
>
> **7. [Ehtical purpose]**
>
> > Comment: Potential negative societal impacts have not been discussed.
>
> Thank you for the insightful comment. Since multiple reviewers have mentioned this point, we added supplementary material that shows all the authors reviewed ethical concerns.
>
> **8. [Negative Social Impact]**
>
> A typical negative societal impact during aerial data collection is capturing sensitive areas, such as military zones and privacy-sensitive areas. We have carefully reviewed this aspect, and we inform you that our flight areas are limited to non-military zones and the people in the dataset are not identifiable.
> - The ethics review sheet in the supplementary material provides details.
> - To comply with the comment, we newly added Lines 275-280.

---

### Official Review · Reviewer_ZR1t · 2023-07-21
**A very impressive FAIR dataset with definite use for the maritime surveillance community.**

**Rating:** 9
**Confidence:** 5

**Strengths:**

The clarity and reproducibility of this work constitute a clear strength. The paper meticulously documents the methods by which its data was captured and processed in a way that clearly adheres to FAIR best practices and expands on this documentation in its corresponding data repository and introductory webpage.

The data itself is definitely useful to the community. Maritime object detection is a massive research sphere but has far too few *annotated* multimodal datasets for ground truth validation of emerging algorithms. The fact that this dataset captures two modes, is annotated, and is easy to use will be a big benefit to the domain.

The algorithm described is interesting mostly from a standpoint of seeing how an existing algorithm pipeline can be adapted to this specific kind of dataset.

**Additional Feedback:**

Whether or not this paper gets accepted to this conference, I definitely plan on using your dataset, it will be really useful for work in this domain.

**Clarity:**

This paper was extremely well written, I found no glaring grammatical errors and found it quite easy to read, well done. I was also impressed with how self contained it was, it didn't overly rely on the appendix to clearly describe its algorithm and dataset.

**Correctness:**

The dataset is soundly constructed and well documented in terms of format and content. The approach to evaluating the algorithm against other algorithms at the end is reasonably justified, and the adaption to the algorithm pipeline itself was sound.

**Documentation:**

As mentioned in the "Strengths" section, the documentation for the dataset in the data descriptor and the dataset itself. The methods are clearly described, as are the actual data formats and dataset structure. My only documentation concern would be that the source code hasn't been linked to the paper yet--that needs to happen by the time of publication for full reproducibility. I also noticed that though you mentioned incremental PCA as part of your algorithm pipeline, you didn't clarify which implementation of incremental pca (it has its own rapidly developing sphere of research). Source code would clarify that.

**Ethics:**

No. The dataset is licensed and doesn't contain personal information. My only potential concern is that the repository they used doesn't seem to assign a persistent identifier (DOI). If it does, I didn't see it.

**Limitations:**

I was definitely impressed with the Author's discussion of the main limitation of this kind of data, namely that HSI has low resolution. They not only discussed the implications of this, but clearly used their experiments to demonstrate that HSI alone cannot compete with optical images. However, they also included experiments that clearly showed that supplementing HSI with optical images does have a positive effect, especially in the presence of sea surface effects. Well discussed.

Of course, one other limitation is that although the methodology is clearly documented and therefore reproducible, it required an actual airplane with the appropriate sensors and is therefore expensive to collect.

**Opportunities For Improvement:**

I was largely satisfied with the paper, which is well contained, but two points jumped out for improvement.
1. I was surprised that multimodal data fusion with SAR was not mentioned explicitly, since it's so integral to the state of the art maritime object detection. I would recommend including at least a sentence or two in the "multi modal data fusion" section of prior work distinguishing why HSI + optical is worthwhile as opposed to SAR+<other sources>.
2. I liked the experimental section and the metrics used, but I would recommend adding a section about the comparative performance times for each algorithm. Maritime surveillance in active scenarios tends to be high risk and correspondingly time sensitive, so performance time is a critical metric.

**Relation To Prior Work:**

To my knowledge, the relation to prior work captured a lot of the salient points for the current state of publicly available datasets for object detection in maritime surveillance. As I mentioned the only gap was not discussing explicitly some of the prior multi-modal data fusion work with SAR because it is so ubiquitous.

**Summary And Contributions:**

This work includes two key contributions:
1. Introducing a new dataset with registered and annotated optical and hyperspectral maritime images.
2. Introducing a new extension of the feature pyramid network designed specifically to handle multi-modal data like the unique M2SODAI dataset introduced.
The power of this work is not only its novel dataset which has clear applications for validating maritime object detection methods, but in the meticulous documentation that could facilitate new, similar datasets being collected.

---

> ### Author Response · Authors · 2023-08-20
> **Response to Reviewer ZR1t**
>
> Thank you sincerely for your thoughtful review, and we apologize for the delayed response. We have prepared point-wise answers to the reviewer's comments as detailed below. We believe that the above discussion and response have greatly enhanced the completeness of the dataset.
>
> **1. [Comparison with multi-modal SAR dataset in related works section]**
>
> > Comment: I was surprised that multimodal data fusion with SAR was not mentioned explicitly, since it's so integral to the state of the art maritime object detection. I would recommend including at least a sentence or two in the "multi modal data fusion" section of prior work distinguishing why HSI + optical is worthwhile as opposed to SAR+<other sources>.
>
> We sincerely thank you for providing new insights into the related works. While SAR data encapsulates information about an object's shape, HSI focuses on the object's spectral information, gathering complementary details from both areas. This leads to the assertion that HSI data may be more suitable for sea surface object detection. A reference has been included to support this.
> - We newly added Lines 523-528 in the supplementary to comply with the comment for the SAR dataset review.
>
> [r1] Spencer Low, Oliver Nina, Angel D. Sappa, Erik Blasch, and Nathan Inkawhich. Multi-modal aerial view object classification challenge results - pbvs 2023. In Proceedings of the IEEE/CVF Conference on Computer Vision and Pattern Recognition (CVPR) Workshops, pages 412–421, June 2023.
>
> **2. [Newly added experimental results on inference time and corresponding discussion]**
>
> >Comment: I liked the experimental section and the metrics used, but I would recommend adding a section about the comparative performance times for each algorithm. Maritime surveillance in active scenarios tends to be high risk and correspondingly time sensitive, so performance time is a critical metric.
>
> As the reviewer has noted, inference time is one of the most critical factors for maritime safety. We have meticulously included it in Table G2, located in the appendix. In accordance with another reviewer, image fusion techniques handling additional HSI tend to display longer delays compared to methods that utilize only RGB. Therefore, we have encouraged future research on a more computationally efficient and effective 3D CNN-based feature mapping for HSI, with a focus on key feature extraction.
> - We revised Table G2 and newly added Lines 768-770 for additional experimental results.
> - We also added a new limitation on the computational complexity in lines 270-274
>
> **3. [Additional limitation on costly data collection]**
>
> > Comment: Of course, one other limitation is that although the methodology is clearly documented and therefore reproducible, it required an actual airplane with the appropriate sensors and is therefore expensive to collect.
>
> The authors believe that there are avenues to enhance the costly data collection method in the future. Additionally, we concur that this represents one of the limitations of our study. We have included this constraint regarding the collection method in the limitations paragraph in Section 5.
>
> - For following the reviewer’s guide, we added Lines 268-270 for a new costly limitation.
>
> **4. [DOI Generation]**
>
> > Comment: The dataset is licensed and doesn't contain personal information. My only potential concern is that the repository they used doesn't seem to assign a persistent identifier (DOI). If it does, I didn't see it.
>
> As the reviewer has pointed out, the DOI of the Dataset is an essential element for ensuring its persistence. To facilitate continuous management and distribution of the dataset, we have created a DOI in the Hugging Face repository.
> - The generated DOI is: 10.57967/hf/0986.

---

### Author Response · Authors · 2023-08-20
**Revised manuscript is uploaded.**

We have uploaded the revised manuscript, which incorporates the feedback from all five reviewers. Please refer to the attached files and the point-by-point response for detailed information.

---

> ### Author Response · Authors · 2023-08-26
>
> Dear reviewers,
>
> A few days ago we submitted a revised manuscript and corresponding responses.
>
> We kindly remind you that the discussion period is up to the 29th.

---

### Author Response · Authors · 2023-08-28

Dear Reviewers and Chairs,

We did a point-by-point response to the reviewers' comments. Since then, the authors have not been able to find further comments.
The reviewer double-checked but found no issues.
Maybe there is something wrong that we don't know about? I'd really appreciate it if you let me know if you find any issues.

---

### Decision · Program_Chairs · 2023-09-22

**Decision:**

Accept (Poster)

**Comment:**

Four out of five reviewers recommend accept of this paper. They find the paper solid on writing, quality of the dataset and experiments. Above it, they find that a maritime object detection a useful application with clear societal impact.
One of the reviewers has a few concerns and recommends reject in the preliminary review. These concerns has been addressed carefully but the authors.
In conclusion this paper is good for acceptance.